# Greener GRASS: Enhancing GNNs with Encoding, Rewiring, and Attention

**Tongzhou Liao** ⦿
School of Computer Science
Carnegie Mellon University
Pittsburgh, USA

**Barnabás Póczos**
School of Computer Science
Carnegie Mellon University
Pittsburgh, USA

## Abstract

Graph Neural Networks (GNNs) have become important tools for machine learning on graph-structured data. In this paper, we explore the synergistic combination of graph encoding, graph rewiring, and graph attention, by introducing Graph Attention with Stochastic Structures (GRASS), a novel GNN architecture. GRASS utilizes relative random walk probabilities (RRWP) encoding and a novel decomposed variant (D-RRWP) to efficiently capture structural information. It rewires the input graph by superimposing a random regular graph to enhance long-range information propagation. It also employs a novel additive attention mechanism tailored for graph-structured data. Our empirical evaluations demonstrate that GRASS achieves state-of-the-art performance on multiple benchmark datasets, including a 20.3% reduction in mean absolute error on the ZINC dataset.

## 1 Introduction

Graph Neural Networks (GNNs) have revolutionized machine learning tasks involving graph-structured data (Wu et al., 2020; Veličković, 2023). Various paradigms within GNNs offer distinct advantages: message-passing neural networks (MPNNs) effectively leverage local graph structure (Kipf and Welling, 2016) and are often complemented with graph rewiring techniques to modify the graph's topology and facilitate message passing (Topping et al., 2021); Graph Transformers (GTs) incorporate attention mechanisms to capture global dependencies (Yun et al., 2019; Rampášek et al., 2022) and can be enhanced by graph encoding methods that enrich node and edge features with structural information (Dwivedi et al., 2021; 2023).

The goal of this work is to create a new architecture that can possess the advantageous properties of the methods discussed above. To achieve this goal, we propose Graph Attention with Stochastic Structures (GRASS), a GNN architecture that synergistically combines random walk encoding, random rewiring, and introduces a novel additive attention mechanism designed for graph-structured data. We conduct a series of experiments on multiple benchmark datasets and perform ablation studies to rigorously assess the contribution of each component in GRASS. Our results show that GRASS achieves competitive or superior performance compared to existing methods on multiple popular datasets, suggesting that the synergy of random walk encoding, random rewiring, and a graph-tailored attention mechanism can effectively enhance GNNs.

**Our Contributions.**

- We propose GRASS, a GNN architecture that integrates random walk encoding, random rewiring, and a novel additive attention mechanism designed for graphs.
- We analyze these components with respect to desirable properties of a GNN and provide insights into how they contribute to the model's performance.
- We provide empirical evidence through experiments and ablation studies that a carefully selected combination of these components can lead to improved performance on multiple benchmark datasets.

In the remainder of the paper, we review related work in Section 2, describe the architecture of GRASS in Section 3, present experimental results in Section 4, and draw conclusions in Section 5.

## 2 RELATED WORK

In this section, we summarize some of the previous work related to the main concepts of GRASS.

**Message-Passing Neural Networks.** Message-Passing Neural Networks (MPNNs), such as Graph Convolutional Networks (GCNs) (Kipf and Welling, 2016), GraphSAGE (Hamilton et al., 2017), and Graph Isomorphism Networks (GIN) (Xu et al., 2018), propagate information within local neighborhoods of a graph. By aligning computation with the structure of the graph, MPNNs offer a strong inductive bias for graph-structured data (Ma et al., 2023).

**Graph Rewiring.** Graph rewiring techniques modify the topology of graphs to improve their connectivity, often leveraging spectral properties to guide the process (Topping et al., 2021; Arnaiz-Rodriguez et al., 2022). Rewiring improves MPNNs by alleviating issues in information propagation, such as underreaching, which occurs when distant nodes cannot communicate (Alon and Yahav, 2020). In this work, we explore a form of random rewiring that superimposes a random regular graph on the input graph.

**Graph Transformers.** Attention mechanisms (Vaswani et al., 2017) allow GNNs to weigh the importance of neighboring nodes during aggregation (Veličković et al., 2017). Graph Transformers (GTs), such as Graph Transformer Network (GTN) (Yun et al., 2019), extend this idea to global attention across nodes. GTs often inherit attention mechanisms designed for sequences, which may not be optimal for graphs (Chen et al., 2024). Designing attention mechanisms specifically for graph-structured data is an active area of research, and we aim to contribute to it by proposing a novel additive attention mechanism.

**Graph Encoding.** Enhancing node and edge features with graph encodings has been shown to improve GNN performance, especially for GTs (Dwivedi et al., 2023). Techniques such as Laplacian positional encodings (LapPE) (Dwivedi et al., 2021) and relative random walk probabilities (RRWP) encoding (Ma et al., 2023) incorporate structural information into node and edge features, enhancing GTs, which otherwise lack a graph inductive bias (Ma et al., 2023). We utilize RRWP encoding in GRASS, and propose a decomposed variant (D-RRWP) with improved computational efficiency.

**Notable Combinations.** The General, Powerful, Scalable (GPS) Graph Transformer (Rampášek et al., 2022) represents a hybrid of MPNN and GT, merging the inductive bias of message passing with the global perspective of Transformers. Exphormer (Shirzad et al., 2023) combines GTs and rewiring by adding random edges and supernodes, generalizing BigBird (Zaheer et al., 2020), a sparse Transformer, to graph-structured data.

Our work builds upon these paradigms by incorporating random walk encoding and random rewiring, and we also create a novel additive attention mechanism tailored for graphs. Although RRWP encoding and random rewiring have been explored separately (Ma et al., 2023; Shirzad et al., 2023), their combination with each other and a graph-tailored attention mechanism is, to the best of our knowledge, novel. Our experiments show that this architecture not only matches but often exceeds state-of-the-art performance across a wide range of benchmark problems.

## 3 METHODS

In this section, we introduce the design of GRASS. We begin by examining the desirable qualities of a GNN, which guide our architectural design. Subsequently, we introduce the components of GRASS by following the order of data processing in our model, and describe the role of each component in terms of the design goals.

**Design Goals.** We center our design around what we consider to be the key characteristics of an effective GNN. We will focus on the processing of nodes (*N1–N3*) and edges (*E1–E2*) of the model.

*N1. Permutation Equivariance.* Unlike tokens in a sentence or pixels in an image, nodes in a graph are unordered, and therefore the model should be permutation equivariant by construction. Since reordering the nodes of a graph does not change the graph, permuting the nodes of the

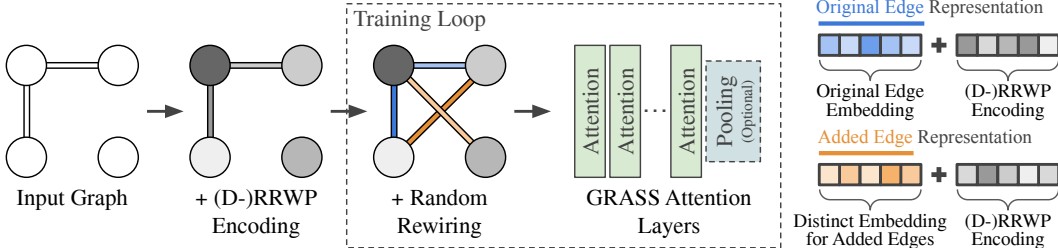

Figure 1: The structure of GRASS. Prior to training, GRASS precomputes (D-)RRWP encodings. At each training iteration, it rewires the input graph and adds distinct embeddings to added edges.

input graph of a GNN layer should only result in the same permutation of its output (Veličković, 2023). Formally, let $f(\mathbf{X}, \mathbf{E}, \mathbf{A})$ be the function computed by a layer, where $\mathbf{X} \in \mathbb{R}^{|V| \times n_{\text{node}}}$ represents node features with $n_{\text{node}}$ dimensions, $\mathbf{E} \in \mathbb{R}^{|V| \times |V| \times n_{\text{edge}}}$ represents edge features with $n_{\text{edge}}$ dimensions, and $\mathbf{A} \in \{0, 1\}^{|V| \times |V|}$ represents the adjacency matrix (edge weights are considered scalar-valued edge features). If the layer is permutation equivariant and $(\mathbf{X}_{\text{out}}, \mathbf{E}_{\text{out}}) = f(\mathbf{X}_{\text{in}}, \mathbf{E}_{\text{in}}, \mathbf{A})$, then $(\mathbf{P}\mathbf{X}_{\text{out}}, \mathbf{P}\mathbf{E}_{\text{out}}\mathbf{P}^\top) = f(\mathbf{P}\mathbf{X}_{\text{in}}, \mathbf{P}\mathbf{E}_{\text{in}}\mathbf{P}^\top, \mathbf{P}\mathbf{A}\mathbf{P}^\top)$ for an arbitrary permutation matrix $\mathbf{P}$.

*N2. Effective Communication* – The model should facilitate long-range communication between nodes. Numerous real-world tasks require the GNN to capture interactions between distant nodes (Dwivedi et al., 2022). However, MPNN layers, which propagate information locally, frequently fail in this regard (Ma et al., 2023). A major challenge is underreaching, where an MPNN with $l$ layers is incapable of supporting communication between two nodes $i, j$ with distance $\delta(i, j) > l$ (Alon and Yahav, 2020). Another challenge is oversquashing, where the structure of the graph forces information from a large set of nodes to squeeze through a small set of nodes to reach its target (Topping et al., 2021). A node with a constant-size feature vector may need to relay information from exponentially many nodes (with respect to model depth), leading to excessive compression of messages in deep MPNNs (Alon and Yahav, 2020).

*N3. Selective Aggregation* – The model should only aggregate information from relevant nodes and edges. MPNN layers commonly update node representations by unconditionally summing or averaging messages from neighboring nodes and edges (Veličković, 2023). In deep models, this can lead to oversmoothing, where the representation of nodes becomes too similar to be effectively classified in the process of repeated aggregation (Chen et al., 2020). Therefore, when required by the task, nodes should aggregate information from relevant neighbors only, instead of doing so unconditionally, in order to maintain distinguishability of node representations.

*E1. Relationship Representation* – The model should effectively represent the relationships between nodes with edges. Edges in graph-structured data often convey meaningful information about the relationship between the nodes they connect (Gong and Cheng, 2019). In addition to the semantic relationships represented by edge features of the input graph, structural relationships can be represented by edge encodings added by the model (Rampášek et al., 2022). To capture the relationships between nodes, edge representations should combine information from both edge features and encodings, and undergo deep processing through multiple layers.

*E2. Directionality Preservation* – The model should preserve and utilize information carried by edge directions. Many graphs representing real-world relationships are inherently directed (Rossi et al., 2024). Although edge directionality has been shown to carry important information for various tasks, many GNN variants require undirected graphs as input, to prevent edge directions from restricting information flow (Rossi et al., 2024). It would be beneficial for the model's expressivity if edge directionality information could be preserved without severely limiting communication.

**The Structure of GRASS.** The high-level structure of GRASS is illustrated in Figure 1. Prior to training, GRASS precomputes the (D-)RRWP encoding of each graph in the dataset. At each training iteration, GRASS randomly rewires the input graph, applies node and edge encodings, and passes the graph through multiple attention layers, producing an output graph with the same structure. For tasks that require a graph-level representation, such as graph regression and graph classification, pooling is performed on the output graph to obtain a single output vector.

### 3.1 GRAPH ENCODING

Extracting structural information plays an important role in graph-structured learning and is crucial for *Relationship Representation*. To this end, we apply relative random walk probabilities (RRWP) encoding (Ma et al., 2023) to represent structural relationships. In addition, we propose D-RRWP, a decomposed variant of RRWP that offers improved computational efficiency.

**RRWP Encoding.** RRWP encoding has been shown to be an expressive representation of graph structure both theoretically and practically (Ma et al., 2023), serving as a major source of structural information for the model. To calculate random walk probabilities, we first obtain the transition matrix $\mathbf{T}$, where $\mathbf{T}_{i,j}$ represents the probability of moving from node $i$ to node $j$ in a random walk step. It is defined as $\mathbf{T} = \mathbf{D}^{-1}\mathbf{A} \in [0,1]^{|V| \times |V|}$, where $\mathbf{A} \in \{0,1\}^{|V| \times |V|}$ is the adjacency matrix of the input graph $G$, and $\mathbf{D} \in \mathbb{N}^{|V| \times |V|}$ is its degree matrix, a diagonal matrix. The powers of $\mathbf{T}$ are stacked to form the RRWP tensor $\mathbf{P}$, with $\mathbf{P}_{h,i,j}$ representing the probability that a random walker who starts at node $i$ lands at node $j$ at the $h$-th step. Formally, $\mathbf{P} = [\mathbf{T}, \mathbf{T}^2, ..., \mathbf{T}^k] \in [0,1]^{k \times |V| \times |V|}$, where $k$ is the number of random walk steps. The diagonal elements $\mathbf{P}_{:,i,i}$, where $i \in V_G$, are used as node encodings, similarly to RWSE (Dwivedi et al., 2021). The rest are used as edge encodings when the corresponding edge is present in the rewired graph $H$. All node and edge encodings undergo batch normalization (BN) (Ioffe and Szegedy, 2015). Here, $\mathbf{W}$ denotes trainable weights, $n$ denotes the dimensionality of hidden layers, and $\|$ denotes concatenation.

$$\mathbf{x}_i^{\text{RW}} = \mathbf{W}_{\text{node-enc}} \, \text{BN}(\mathbf{P}_{:,i,i}) \in \mathbb{R}^n \,, \tag{1}$$

$$\mathbf{e}_{i,j}^{\text{RW}} = \mathbf{W}_{\text{edge-enc}} \, \text{BN}(\mathbf{P}_{:,i,j} \, \| \, \mathbf{P}_{:,j,i}) \in \mathbb{R}^n \,. \tag{2}$$

Before entering attention layers, RRWP encodings are added to both node features and edge features, including those of edges added by random rewiring. The node encodings are additionally accompanied by degree information (Ying et al., 2021). Here, $\text{d}^+(i)$ and $\text{d}^-(i)$ denote the out-degree and in-degree of node $i$, respectively.

$$\mathbf{x}_i^0 = \mathbf{x}_i^{\text{in}} + \mathbf{x}_i^{\text{RW}} + \mathbf{W}_{\text{deg}} \, \text{BN}(\text{d}^+(i) \, \| \, \text{d}^-(i)) \in \mathbb{R}^n \,, \tag{3}$$

$$\mathbf{e}_{i,j}^0 = \mathbf{e}_{i,j}^{\text{in}} + \mathbf{e}_{i,j}^{\text{RW}} \in \mathbb{R}^n \,. \tag{4}$$

**Improving Efficiency.** RRWP encodings take $O(k|V||E|)$ time to compute and $O(k|V|^2)$ space to store (Ma et al., 2023). On extremely large graphs, this could be computationally prohibitive even when computed once per dataset. We propose D-RRWP, a decomposed variant of RRWP. Instead of computing the exact random walk probabilities $\mathbf{P}$ from the transition matrix $\mathbf{T}$, we approximate it with its truncated eigendecomposition to reduce the complexity to $O(km(|V| + |E|))$ time and $O((k+m)|V| + k|E|)$ space, where $m \leq |V|$ is the number of eigenpairs used for the approximation.

To ensure that the transition matrix is diagonalizable, we replace $\mathbf{T}$ with $\mathbf{T}_{\text{sym}} = \mathbf{D}^{-\frac{1}{2}}\mathbf{A}\mathbf{D}^{-\frac{1}{2}}$, which is always diagonalizable if $\mathbf{A}$ is a symmetric adjacency matrix—that of an undirected graph. Given the degree matrix $\mathbf{D}$, this modification does not result in a loss of information, because $\mathbf{T}_{\text{sym}} = \mathbf{D}^{\frac{1}{2}}\mathbf{T}\mathbf{D}^{-\frac{1}{2}}$. Since $\mathbf{T}_{\text{sym}}$ is diagonalizable, its truncated eigendecomposition coincides with its truncated singular value decomposition (SVD), which provides the optimal low-rank approximation of the matrix (Eckart and Young, 1936).

Let $\tilde{\mathbf{T}}_{\text{sym}} = \tilde{\mathbf{Q}}\tilde{\mathbf{\Lambda}}\tilde{\mathbf{Q}}^\top$ be the truncated eigendecomposition of $\mathbf{T}_{\text{sym}}$, where $\tilde{\mathbf{\Lambda}} \in [-1,1]^{m \times m}$ is a diagonal matrix containing the $m$ largest eigenvalues (in magnitude) of $\mathbf{T}_{\text{sym}}$ and $\tilde{\mathbf{Q}} \in \mathbb{R}^{|V| \times m}$ hold the corresponding eigenvectors. Decomposing $\mathbf{T}_{\text{sym}}$ takes $O(m|E|)$ time with the Lanczos algorithm (Lanczos, 1950; Lehoucq et al., 1998) and requires $O(m|V|)$ space. We approximate the $h$-th power of $\mathbf{T}_{\text{sym}}$, i.e. the random walk probabilities $\mathbf{P}_{h,i,j} = (\mathbf{T}_{\text{sym}}^h)_{i,j}$, by computing $(\tilde{\mathbf{T}}_{\text{sym}}^h)_{i,j} = (\tilde{\mathbf{Q}}\tilde{\mathbf{\Lambda}}^h\tilde{\mathbf{Q}}^\top)_{i,j} = (\tilde{\mathbf{Q}}_{i,:} \odot \tilde{\mathbf{Q}}_{j,:}) \cdot \text{diag}(\tilde{\mathbf{\Lambda}}^h)$. Here, $\odot$ denotes the Hadamard product. Computing this approximation for all nodes $(i,i)$, edges $(i,j)$, and random walk steps $1 \leq h \leq k$ requires $O(km(|V| + |E|))$ time and $O(k(|V| + |E|))$ space. This is efficient because raising the diagonal matrix $\tilde{\mathbf{\Lambda}}$ to the powers 1 to $k$ only costs $O(km)$ time and space. If $\mathbf{T}_{\text{sym}}$ were not diagonalizable and we had to use SVD instead, efficient power computation would no longer be possible.

Since $\tilde{\mathbf{\Lambda}}^h$ can take at most $m$ linearly independent values as we vary $h$, increasing $k$ beyond $m$ does not add additional information to the encoding. Consequently, we fix $k = m$ in our experiments.

## 3.2 RANDOM REWIRING

To achieve *Effective Communication*, GRASS rewires the input graph by superimposing a random regular graph. We present some motivations for using random regular graphs instead of deterministic or non-regular graphs in Appendix A.1. Shirzad et al. (2023) uses a similar technique in generalizing BigBird (Zaheer et al., 2020) to graphs, and discusses additional motivations. We also demonstrate the advantage of using random regular graphs with empirical results shown in Figure 4 and Table 3. Here, we provide details on random regular graph generation and input graph rewiring.

**Generating Random Regular Graphs.** We generate random regular graphs with the Permutation Model (Friedman et al., 1989) that we describe here and with pseudocode in Appendix A.2. For a given positive, even parameter $r \geq 2$, and for the input graph $G = (V_G, E_G)$, we randomly generate a corresponding $r$-regular graph by independently and uniformly sampling $\frac{r}{2}$ random permutations $\sigma_1, \sigma_2, ..., \sigma_{\frac{r}{2}}$ from $S_{|V_G|}$, the symmetric group defined over the nodes of the graph $G$. Using these random permutations, we construct a random pseudograph $\tilde{R} = (V_G, E_{\tilde{R}})$, where the edge set $E_{\tilde{R}}$ of the graph $\tilde{R}$ is

$$E_{\tilde{R}} = \bigsqcup_{i \in V_G, \, j=1,...,\frac{r}{2}} \{\{i, \sigma_j(i)\}\}. \tag{5}$$

Here, $\sqcup$ denotes the disjoint union of sets. The resulting graph $\tilde{R}$ is a random regular pseudograph, and the probability that $\tilde{R}$ is any given regular pseudograph with $|V_G|$ nodes and degree $r$ is uniform (Friedman et al., 1989).

Being a pseudograph, $\tilde{R}$ might not be simple—it might contain self-loops and multi-edges. Even when $|V_G|$ is large, the probability that $\tilde{R}$ is simple—that it does not contain self-loops or multi-edges—would not be prohibitively small. In particular, it has an asymptotically tight lower bound (Ellis, 2011)

$$\lim_{|V_G| \to \infty} \Pr[\tilde{R} \text{ is simple}] = e^{-\left(\frac{r^2}{2} + r\right)}. \tag{6}$$

Therefore, if we regenerate $\tilde{R}$ when it is not simple, the expected number of trials required for us to obtain a simple $\tilde{R}$ is upper-bounded by $e^{\frac{r^2}{2} + r}$, which can be kept low by keeping $r$ low. In practice, GRASS requires the superimposed graph to be simple to avoid passing duplicate messages. Meanwhile, the regularity of the graph is desired but not strictly required. Therefore, when $\tilde{R}$ is not simple, we remove self-loops and multi-edges from $\tilde{R}$ to obtain $R$, which is always simple but not necessarily regular.

Our empirical results presented in Figure 4 and Table 3 suggest that a small value of $r$ is often sufficient. In our experiments, we use $r \leq 6$.

**Rewiring the Input Graph.** To rewire the input graph $G$, GRASS superimposes the edges of $R$ on $G$, producing a new graph $H = (V_G, E_G \sqcup E_R)$ that is used as input for subsequent stages of the model. Since it is possible that $E_G \cap E_R \neq \varnothing$, there might be multi-edges in $H$, and in these cases, $H$ is not a simple graph. GRASS does not remove these multi-edges to avoid biasing the distribution of the superimposed random regular graph.

As illustrated in Figure 1, the added edges $E_R$ are given a distinct embedding to distinguish them from the existing edges $E_G$. This aids *Selective Aggregation*, as it allows a node to select between its neighbors and a random node. The added edges also receive (D-)RRWP encodings to enhance *Relationship Representation*. Although an added edge $(i, j) \in E_R$ lacks edge features that represent semantic relationships in the input graph, the structural relationship between nodes $i$ and $j$ can be represented by the random walk probabilities $\mathbf{P}_{:,i,j}$ given to that edge as its (D-)RRWP encoding.

## 3.3 ATTENTION MECHANISM

Many GTs emulate the structure of Transformers designed for Euclidean data (Dwivedi and Bresson, 2020; Kreuzer et al., 2021; Ying et al., 2021; Hussain et al., 2022; Shirzad et al., 2023). Meanwhile, GRASS uses attentive node aggregators with attention scores computed by MLP edge aggregators, which is a more tailored attention mechanism for graph-structured data. Figure 3 provides a simple

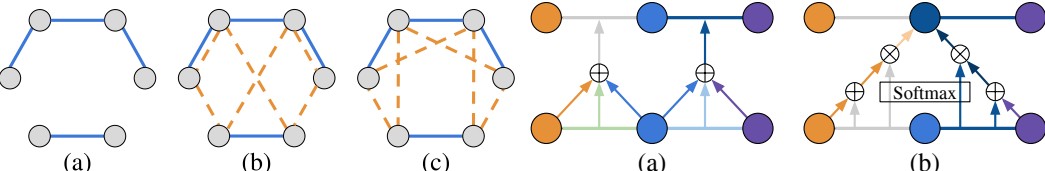

Figure 2: Visualization of the proposed random rewiring technique. Solid lines denote existing edges of the input graph, and dashed lines denote added edges. **(a)** An example of the input graph $G$ that has poor connectivity. **(b, c)** Two among all possible instances of the randomly rewired graph $H$ with $r = 2$. They have better connectivity than the input graph.

Figure 3: Simplified visualization of the GRASS attention mechanism. **(a)** The edge aggregator extracting node relations to update edge representations. **(b)** The attentive node aggregator weighted by edge representations. For simplicity, attention from a node to itself, residual connections, and activation functions are omitted here. Figure 5 provides a more detailed visualization.

visualization, and Figure 5 illustrates its structure in detail. The GRASS attention mechanism is defined as follows, where $\mathcal{N}^-$ denotes in-neighbors, $\mathbf{W}$ denotes trainable weights, and $\varepsilon$ denotes a small constant added for numerical stability. For simplicity, biases are not shown in these equations.

$$\mathbf{s}_{i,j}^l = \text{dropout}(\exp(\mathbf{W}_{\text{attn} \leftarrow \text{edge}}^l \mathbf{e}_{i,j}^{l-1})) \in \mathbb{R}^{+n}\,, \tag{7}$$

$$\mathbf{a}_{i,j}^l = \frac{\mathbf{s}_{i,j}^l}{\sum_{h \in \mathcal{N}^-(j) \cup \{j\}} \mathbf{s}_{h,j}^l + \varepsilon} \in \mathbb{R}^{+n}\,, \tag{8}$$

$$\tilde{\mathbf{x}}_j^l = \mathbf{W}_{\text{tail} \leftarrow \text{tail}}^l \mathbf{x}_j^{l-1} + \sum_{i \in \mathcal{N}^-(j) \cup \{j\}} \mathbf{a}_{i,j}^l \odot (\mathbf{W}_{\text{tail} \leftarrow \text{head}}^l \mathbf{x}_i^{l-1} + \mathbf{W}_{\text{tail} \leftarrow \text{edge}}^l \mathbf{e}_{i,j}^{l-1}) \in \mathbb{R}^n\,, \tag{9}$$

$$\tilde{\mathbf{e}}_{j,i}^l = \mathbf{W}_{\text{edge} \leftarrow \text{edge}}^l \mathbf{e}_{i,j}^{l-1} + \mathbf{W}_{\text{edge} \leftarrow \text{head}}^l \mathbf{x}_i^{l-1} + \mathbf{W}_{\text{edge} \leftarrow \text{tail}}^l \mathbf{x}_j^{l-1} \in \mathbb{R}^n\,. \tag{10}$$

**Attention Weights.** This attention mechanism is unique in the way it uses edge representations as the medium of attention weights. To achieve *Relationship Representation*, edge representations must be updated alongside node representations (Zhou et al., 2023). A directed edge is an ordered pair of nodes, and an undirected edge can be represented by two directed edges with opposite directions. The orderedness of nodes connected by an edge allows us to use an MLP as the edge aggregator while preserving *Permutation Equivariance*. Assuming that *Relationship Representation* is satisfied, edge features would already represent node relationships, which could be used as attention weights $\mathbf{a}_{i,j}$ after applying a linear layer and performing a component-wise softmax over the neighborhood.

**Random Edge Removal.** To complement the proposed random rewiring technique, which adds edges but never removes them, GRASS attention randomly removes edges in computing attention weights with the dropout function in Equation 7. The goal is to reduce the model's dependence on the presence of particular edges in the graph, facilitating *Selective Aggregation*. This can also be seen as a generalization of DropKey (Li et al., 2022) to graphs, because it randomly masks the attention matrix prior to the softmax operation, unlike DropAttention (Zehui et al., 2019).

**Edge Flipping.** While the proposed attention mechanism naturally achieves *Directionality Preservation* by aggregating information in the same direction as edges, it can severely restrict the flow of information, putting it in conflict with *Effective Communication*. As a solution, the direction of each edge is switched from one layer to the next: in odd-numbered layers, the directions match those of the edges in the rewired graph $H$, while in even-numbered layers, they are reversed. This enables the model to propagate information in both directions even when the input graph is directed, improving its expressivity. In Equation 10, we compute $\tilde{\mathbf{e}}_{j,i}^l$, the updated representation of edge $j \rightarrow i$ by using $\mathbf{e}_{i,j}^{l-1}$, the representation of edge $i \rightarrow j$ from the previous layer, effectively flipping the edge.

**Feed-Forward Network.** Similar to Transformers, the output of the attention mechanism is passed through an FFN. Here, $\phi$ denotes a ReLU-like (Nair and Hinton, 2010; Ramachandran et al., 2017) nonlinear activation function, which we choose to be Mish (Misra, 2019).

$$\hat{\mathbf{x}}_i^l = \mathbf{W}_{\text{node-out}}^l \phi(\tilde{\mathbf{x}}_i^l + \mathbf{b}_{\text{node-act}}^l) + \mathbf{b}_{\text{node-out}}^l \in \mathbb{R}^n\,, \tag{11}$$

$$\hat{\mathbf{e}}_{i,j}^l = \mathbf{W}_{\text{edge-out}}^l \phi(\tilde{\mathbf{e}}_{i,j}^l + \mathbf{b}_{\text{edge-act}}^l) + \mathbf{b}_{\text{edge-out}}^l \in \mathbb{R}^n\,. \tag{12}$$

Table 1: Performance on GNN Benchmark Datasets. The performance of GRASS shown here is the mean ± s.d. of 16 runs on ZINC, and 8 runs on other datasets. The **best** and second-best results are highlighted. Performance numbers other than that of GRASS are adapted from Ma et al. (2023), Shirzad et al. (2023), and Chen et al. (2024). "-" indicates experiments not reported in these works.

| Model | ZINC MAE ↓ | MNIST Accuracy ↑ | CIFAR10 Accuracy ↑ | PATTERN Accuracy ↑ | CLUSTER Accuracy ↑ |
|---|---|---|---|---|---|
| GCN | 0.367 ± 0.011 | 90.705 ± 0.218 | 55.710 ± 0.381 | 71.892 ± 0.334 | 68.498 ± 0.976 |
| GIN | 0.526 ± 0.051 | 96.485 ± 0.252 | 55.255 ± 1.527 | 85.387 ± 0.136 | 64.716 ± 1.553 |
| GAT | 0.384 ± 0.007 | 95.535 ± 0.205 | 64.223 ± 0.455 | 78.271 ± 0.186 | 70.587 ± 0.447 |
| GatedGCN | 0.282 ± 0.015 | 97.340 ± 0.143 | 67.312 ± 0.311 | 85.568 ± 0.088 | 73.840 ± 0.326 |
| GatedGCN-LSPE | 0.090 ± 0.001 | - | - | - | - |
| PNA | 0.188 ± 0.004 | 97.94 ± 0.12 | 70.35 ± 0.63 | - | - |
| DGN | 0.168 ± 0.003 | - | 72.838 ± 0.417 | 86.680 ± 0.034 | - |
| GSN | 0.101 ± 0.010 | - | - | - | - |
| CIN | 0.079 ± 0.006 | - | - | - | - |
| CRaW1 | 0.085 ± 0.004 | 97.944 ± 0.050 | 69.013 ± 0.259 | - | - |
| GIN-AK+ | 0.080 ± 0.001 | - | 72.19 ± 0.13 | 86.850 ± 0.057 | - |
| SAN | 0.139 ± 0.006 | - | - | - | 76.691 ± 0.65 |
| Graphormer | 0.122 ± 0.006 | - | - | - | - |
| K-Subgraph SAT | 0.094 ± 0.008 | - | - | 86.848 ± 0.037 | 77.856 ± 0.104 |
| EGT | 0.108 ± 0.009 | 98.173 ± 0.087 | 68.702 ± 0.409 | 86.821 ± 0.020 | 79.232 ± 0.348 |
| Graphormer-URPE | 0.086 ± 0.007 | - | - | - | - |
| Graphormer-GD | 0.081 ± 0.009 | - | - | - | - |
| GPS | 0.070 ± 0.004 | - | 72.298 ± 0.356 | 86.685 ± 0.059 | 78.016 ± 0.180 |
| Exphormer | - | 98.55 ± 0.039 | 74.69 ± 0.125 | 86.74 ± 0.015 | 78.07 ± 0.037 |
| GRIT | **0.059 ± 0.002** | 98.108 ± 0.111 | 76.468 ± 0.881 | **87.196 ± 0.076** | **80.026 ± 0.277** |
| NeuralWalker | 0.065 ± 0.001 | **98.760 ± 0.079** | **80.027 ± 0.185** | 86.977 ± 0.012 | 78.189 ± 0.188 |
| GRASS (ours) | **0.047 ± 0.001** | **98.932 ± 0.076** | **83.750 ± 0.141** | **89.177 ± 0.313** | **79.541 ± 0.173** |

**Normalization and Residual Connection.** We use post-normalization in residual connections, which has been shown to improve the expressiveness of Transformers (Liu et al., 2020). The residual connection is scaled by a constant $\alpha$ to improve training stability (Wang et al., 2022).

$$\mathbf{x}_i^l = \mathrm{BN}(\mathbf{x}_i^{l-1} + \alpha \hat{\mathbf{x}}_i^l) \in \mathbb{R}^n , \tag{13}$$

$$\mathbf{e}_{i,j}^l = \mathrm{BN}(\mathbf{e}_{i,j}^{l-1} + \alpha \hat{\mathbf{e}}_{i,j}^l) \in \mathbb{R}^n . \tag{14}$$

**Graph Pooling.** For graph-level tasks, graph pooling is required at the output of a GNN to produce a vector representation of each graph, capturing global properties relevant to the task (Wu et al., 2020). GRASS employs sum pooling—a simple method as expressive as the Weisfeiler-Lehman graph isomorphism test (Leman and Weisfeiler, 1968)—while many more complicated pooling methods are not as expressive (Baek et al., 2021). Considering the randomness of the added edges, they are pooled separately from the preexisting edges in the input graph $G$, because the pooled output of the randomly added edges may exhibit a different distribution than that of the preexisting edges. Here, $\|$ denotes concatenation.

$$\mathbf{y} = \sum_{i \in V_G} \mathbf{x}_i^L \left\| \sum_{(i,j) \in E_G} \mathbf{e}_{i,j}^L \right\| \sum_{(i,j) \in E_R} \mathbf{e}_{i,j}^L \in \mathbb{R}^{3n} . \tag{15}$$

## 3.4 INTERPRETATIONS OF GRASS

**A Message Passing Perspective.** GRASS is an MPNN on a noisy graph. In an MPNN, information is propagated along the edges of the input graph, defined by its adjacency matrix (Veličković, 2023). GRASS can be seen as an MPNN that injects additive and multiplicative noise into the adjacency matrix, through random rewiring and random edge removal, respectively. The adjacency matrix $\mathbf{A}_M$ followed by message passing is given by

$$\mathbf{A}_M = (\mathbf{A}_G + \mathbf{A}_R) \cdot \mathbf{A}_D , \tag{16}$$

where $\mathbf{A}_G$ is the adjacency matrix of the input graph $G$, $\mathbf{A}_R$ is that of the superimposed random regular graph $R$, $\mathbf{A}_D$ is a random attention mask sampled by the dropout function in Equation 7 (which can also be seen as the adjacency matrix of a random graph), $+$ denotes element-wise OR, and $\cdot$ denotes element-wise AND. Noise injection is well known as an effective regularizer (Noh et al., 2017), and for graph-structured data, the random removal of edges has demonstrated regularization effects (Rong et al., 2019).

Table 2: Performance on LRGB datasets. The performance of GRASS shown here is the mean ± s.d. of 8 runs. The **best** and **second-best** results are highlighted. Performance numbers other than that of GRASS are adapted from Gutteridge et al. (2023), Tönshoff et al. (2023), Ma et al. (2023), Shirzad et al. (2023), and Chen et al. (2024). "-" indicates experiments not reported in these works. [*]These models are re-tuned by Tönshoff et al. (2023) to provide stronger baselines.

| Model | Peptides-func AP ↑ | Peptides-struct MAE ↓ | PascalVOC-SP Macro F1 ↑ | COCO-SP Macro F1 ↑ |
|---|---|---|---|---|
| GCN[*] | 0.6860 ± 0.0050 | **0.2460 ± 0.0007** | 0.2078 ± 0.0031 | 0.1338 ± 0.0007 |
| GINE[*] | 0.6621 ± 0.0067 | 0.2473 ± 0.0017 | 0.2718 ± 0.0054 | 0.2125 ± 0.0009 |
| GatedGCN[*] | 0.6765 ± 0.0047 | 0.2477 ± 0.0009 | 0.3880 ± 0.0040 | 0.2922 ± 0.0018 |
| DIGL+MPNN | 0.6469 ± 0.0019 | 0.3173 ± 0.0007 | 0.2824 ± 0.0039 | - |
| DIGL+MPNN+LapPE | 0.6830 ± 0.0026 | 0.2616 ± 0.0018 | 0.2921 ± 0.0038 | - |
| MixHop-GCN | 0.6592 ± 0.0036 | 0.2921 ± 0.0023 | 0.2506 ± 0.0133 | - |
| MixHop-GCN+LapPE | 0.6843 ± 0.0049 | 0.2614 ± 0.0023 | 0.2218 ± 0.0174 | - |
| DRew-GCN | 0.6996 ± 0.0076 | 0.2781 ± 0.0028 | 0.1848 ± 0.0107 | - |
| DRew-GCN+LapPE | **0.7150 ± 0.0044** | 0.2536 ± 0.0015 | 0.1851 ± 0.0092 | - |
| DRew-GIN | 0.6940 ± 0.0074 | 0.2799 ± 0.0016 | 0.2719 ± 0.0043 | - |
| DRew-GIN+LapPE | **0.7126 ± 0.0045** | 0.2606 ± 0.0014 | 0.2692 ± 0.0059 | - |
| DRew-GatedGCN | 0.6733 ± 0.0094 | 0.2699 ± 0.0018 | 0.3214 ± 0.0021 | - |
| DRew-GatedGCN+LapPE | 0.6977 ± 0.0026 | 0.2539 ± 0.0007 | 0.3314 ± 0.0024 | - |
| Transformer+LapPE | 0.6326 ± 0.0126 | 0.2529 ± 0.0016 | 0.2694 ± 0.0098 | 0.2618 ± 0.0031 |
| SAN+LapPE | 0.6384 ± 0.0121 | 0.2683 ± 0.0043 | 0.3230 ± 0.0039 | 0.2592 ± 0.0158 |
| GPS+LapPE[*] | 0.6534 ± 0.0091 | 0.2509 ± 0.0014 | 0.4440 ± 0.0065 | 0.3884 ± 0.0055 |
| Exphormer | 0.6527 ± 0.0043 | 0.2481 ± 0.0007 | 0.3975 ± 0.0037 | 0.3455 ± 0.0009 |
| GRIT | 0.6988 ± 0.0082 | **0.2460 ± 0.0012** | - | - |
| NeuralWalker | 0.7096 ± 0.0078 | 0.2463 ± 0.0005 | **0.4912 ± 0.0042** | **0.4398 ± 0.0033** |
| GRASS (ours) | 0.6737 ± 0.0064 | **0.2459 ± 0.0007** | **0.5670 ± 0.0049** | **0.4752 ± 0.0032** |

**A Graph Transformer Perspective.** GRASS is a sparse Graph Transformer. Graph Transformers allow each node to aggregate information from other nodes through graph attention mechanisms, with a general definition (Veličković, 2023) being

$$\mathbf{x}'_j = \phi\left(\mathbf{x}_j, \bigoplus_{i \in \mathcal{N}(j)} a(\mathbf{x}_i, \mathbf{x}_j)\psi(\mathbf{x}_i)\right), \tag{17}$$

where $\phi$ and $\psi$ are neural networks, $a$ is an attention weight function, and $\bigoplus$ is a permutation-invariant aggregator. Many GTs compute attention weights using scaled dot-product attention (Vaswani et al., 2017), with node features as keys and queries. However, we observe that edge features in GRASS, which are updated by aggregating information from its head and tail nodes with an MLP, could be used to directly compute attention weights as a form of additive attention (Bahdanau et al., 2014). *Relationship Representation* would then be critical for the attention weights to be meaningful, which GRASS satisfies through expressive edge encodings and the deep processing of edge features. Many GTs achieve sparsity by integrating (Rampášek et al., 2022) or generalizing (Shirzad et al., 2023) BigBird's sparse dot-product attention. Meanwhile, GRASS achieves sparsity in a graph-native way: attention is always local, so non-adjacent nodes in the rewired graph would naturally never attend to each other. Seeing GRASS as a Transformer, its attention mask would be $\mathbf{A}_M$ as defined in Equation 16, which contains $O(r|V| + |E|)$ nonzero elements.

## 4 EXPERIMENTS

### 4.1 BENCHMARKING GRASS

**Experimental Setup.** To measure the performance of GRASS, we train and evaluate it on five of the GNN Benchmark Datasets (Dwivedi et al., 2023): ZINC, MNIST, CIFAR10, CLUSTER, and PATTERN, as well as four of the Long Range Graph Benchmark (LRGB) (Dwivedi et al., 2022) datasets: Peptides-func, Peptides-struct, PascalVOC-SP, and COCO-SP. Following the experimental setup of Rampášek et al. (2022) and other work that we compare, we configure GRASS to around 100k parameters for MNIST and CIFAR10, and 500k parameters for all other datasets. Additional information on the datasets can be found in Appendix F.1.

Due to the use of random rewiring, the output of the model is not deterministic. Therefore, we evaluate the trained model 100 times for ZINC, and 10 times for other datasets, for each training run. The average performance is reported as the performance of that run. We use D-RRWP encoding

Table 3: Ablation study results for the number of added edges per node, with random regular and non-regular graphs, on ZINC. Reported values for ablated models, except peak VRAM consumption, are the mean ± s.d. over 8 runs. The variance of model performance due to random rewiring are measured by evaluating the test set 100 times on each trained model. For comparison, the variance of model performance due to randomness in the training process is 1.79e-6.

| # Added Edge per Node | | 0 | 3 | 6 | 9 | 12 | Fully Connected |
|---|---|---|---|---|---|---|---|
| MAE ↓ | Regular | 0.0557 ± 0.0021 | 0.0480 ± 0.0019 | 0.0470 ± 0.0013 | 0.0484 ± 0.0018 | 0.0483 ± 0.0012 | 0.0492 ± 0.0008 |
| | Non-Regular | | 0.0488 ± 0.0015 | 0.0486 ± 0.0020 | 0.0480 ± 0.0019 | 0.0475 ± 0.0021 | |
| Variance in MAE Due to Random Rewiring | Regular | Deterministic | 7.60e-8 ± 3.67e-8 | 1.37e-7 ± 1.00e-7 | 1.67e-7 ± 8.51e-8 | 2.24e-7 ± 1.11e-7 | Deterministic |
| | Non-Regular | | 1.20e-7 ± 9.23e-8 | 1.39e-7 ± 4.75e-8 | 1.63e-7 ± 1.19e-7 | 9.94e-8 ± 4.47e-8 | |
| Training Time per Epoch (s) | Regular | 1.74 ± 0.02 | 1.81 ± 0.10 | 1.95 ± 0.07 | 2.02 ± 0.05 | 2.19 ± 0.05 | 2.40 ± 0.03 |
| | Non-Regular | | 1.75 ± 0.04 | 1.87 ± 0.07 | 2.00 ± 0.03 | 2.20 ± 0.04 | |
| Peak VRAM Consumption (MiB) | Regular | 1415 | 1911 | 2529 | 3051 | 3569 | 4273 |
| | Non-Regular | | 1889 | 2489 | 3009 | 3511 | |

on Peptides-func, PascalVOC-SP and COCO-SP, and RRWP encoding on other datasets. Models are trained with the Lion optimizer (Chen et al., 2023). Hyperparameters can be found in Appendix F.2.

**Results.** As shown in Tables 1 and 2, GRASS ranks first on ZINC, MNIST, CIFAR10, PATTERN, Peptides-struct, PascalVOC-SP, and COCO-SP, while ranking second on CLUSTER and fifth on Peptides-func, among the models compared. Notably, GRASS achieves 20.3% lower MAE on ZINC compared to GRIT (Ma et al., 2023), the second-best model, which has $O(|V|^2)$ time and space complexity.

## 4.2 ABLATION STUDY

**Experimental Setup.** We examine the impact of RRWP encoding and D-RRWP encoding by comparing their performance with each other and with LapPE (Dwivedi et al., 2021), a widely used graph encoding technique. We examine the effects of random rewiring by varying the number of added edges per node, and superimposing random non-regular graphs instead of random regular graphs. Furthermore, we assess the effects of

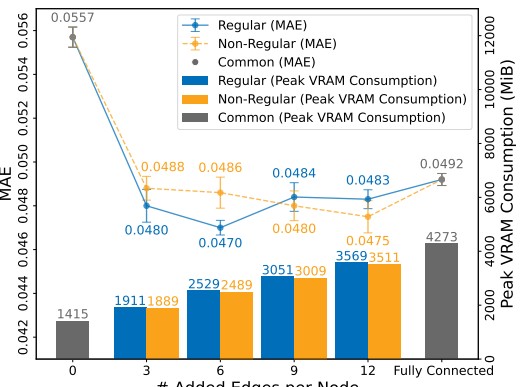

Figure 4: Visualized ablation study results for the number of added edges per node, with random regular and non-regular graphs, on ZINC. Error bars represent one standard error of the mean. The setup is identical to that described in Table 3.

the GRASS attention mechanism by replacing it with the attention mechanisms of GAT (Veličković et al., 2017), GatedGCN (Bresson and Laurent, 2017), and Transformer (Vaswani et al., 2017), while keeping the rest of the model intact. These experiments are conducted on ZINC, a well-known GNN benchmark that represents tasks on smaller graphs, and PascalVOC-SP, which represents tasks on larger graphs that require long-range interaction. Detailed results on PascalVOC-SP are presented in Appendix D. Additionally on ZINC, we explore the impact of minor design choices, including random edge removal, edge flipping, normalization, graph pooling, and the optimizer. Our findings suggest that the combination of (D-)RRWP encoding, random rewiring, and GRASS attention demonstrates superior effectiveness compared to alternative combinations on the evaluated datasets.

**Random Walk Encoding.** On both ZINC and PascalVOC-SP, switching between RRWP and D-RRWP results in an insignificant change in performance: 0.64% on ZINC and 0.35% on PascalVOC-SP. Meanwhile, replacing D-RRWP with RRWP results in a 17.73× larger preprocessed dataset on PascalVOC-SP. This highlights the viability of D-RRWP as an efficient replacement of RRWP on certain graphs, but we also notice that D-RRWP does not perform as well as RRWP on Peptides-struct. Replacing RRWP or D-RRWP with LapPE results in a significant degradation of performance on ZINC, but a much smaller degradation on PascalVOC-SP, indicating that the combination of LapPE with other components of GRASS is more dataset-dependent and generally not as effective.

Table 4: Ablation study results for random rewiring, graph encoding, attention mechanism, and minor design decisions on ZINC. This table shows the performance of each ablated model as the mean ± s.d. over 8 runs. The implementations of replacement attention mechanisms are provided by PyTorch Geometric (Fey and Lenssen, 2019), and we adjust the head size to approach 500k parameters. *The maximum number of Laplacian eigenvectors to use for LapPE on ZINC is 8, which is constrained by the smallest graph in the dataset. For a fair comparison, we include a setup that pads the LapPE of smaller graphs with zeros to raise the maximum number of eigenvectors to 32. †The learning rate is adjusted for these configurations to stabilize training. ‡The batch size, learning rate, betas, and weight decay factor are adjusted for this configuration to stabilize training.

| Setup | MAE ↓ |
|---|---|
| GRASS | 0.0470 ± 0.0013 |
| Rewire at every epoch → Rewire once before training | 0.0645 ± 0.0015 |
| RRWP (32 steps) → D-RRWP (32 eigenpairs, 32 steps) | 0.0473 ± 0.0021 |
| RRWP (32 steps) → LapPE (8 eigenvectors)* | 0.0829 ± 0.0041 |
| RRWP (32 steps) → Padded LapPE (32 eigenvectors)* | 0.0879 ± 0.0067 |
| GRASS attention → GAT attention (Veličković et al., 2017) | 0.0592 ± 0.0023 |
| GRASS attention → GatedGCN attention† (Bresson and Laurent, 2017) | 0.0651 ± 0.0030 |
| GRASS attention → Transformer attention† (Vaswani et al., 2017) | 0.0652 ± 0.0016 |
| No random edge removal | 0.0500 ± 0.0018 |
| No edge flipping | 0.0470 ± 0.0010 |
| BN → LN (Ba et al., 2016) | 0.0497 ± 0.0009 |
| Sum pooling → Mean pooling | 0.0493 ± 0.0024 |
| Lion → AdamW‡ (Loshchilov et al., 2017) | 0.0499 ± 0.0006 |

**Random Rewiring.** On both ZINC and PascalVOC-SP, the optimal number of added edges per node is 6 when using random regular graphs, with any deviation from this value leading to degraded performance. On ZINC, replacing random regular graphs with non-regular random graphs requires adding more edges to achieve comparable performance, which in turn increases runtime and memory consumption. Moreover, fixing the added edges across epochs rather than resampling them at every epoch results in a 37.2% increase in MAE. These findings demonstrate that both the regularity and the randomness of the superimposed graphs are crucial for the model's efficiency and performance.

**GRASS Attention.** On both ZINC and PascalVOC-SP, replacing GRASS attention with alternative attention mechanisms substantially degrades performance, by at least 26.0% on ZINC and 14.7% on PascalVOC-SP. This indicates that GRASS attention, our novel design, is a vital component for GRASS to achieve its competitive performance.

**Minor Design Decisions.** None of the minor design decisions, when altered or removed, results in a significant performance degradation on ZINC: an advantage of at least 15.2% is maintained compared to GRIT, the second-best model. This verifies that the performance advantage of GRASS is achieved mainly by the proposed combination of encoding, rewiring, and attention mechanism.

## 5 CONCLUSION

We have presented GRASS, a novel GNN architecture that synergistically integrates (D-)RRWP encoding, random rewiring, and a new graph-tailored additive attention mechanism. Our empirical evaluations show that GRASS achieves and often surpasses state-of-the-art performance across a diverse set of benchmark problems.

### 5.1 LIMITATIONS

**Empirical Evaluation of Scalability.** GRASS has $O(|V|+|E|)$ time and space complexity, which implies good scalability to large and sparse graphs. While evaluating GRASS on extremely large graphs could provide additional insights, it is beyond the scope of this work due to time constraints.

**Nondeterministic Output.** The output of GRASS is inherently random due to random rewiring. The relationship between performance variance and the number of randomly added edges is demonstrated in Table 3 and Table 6. In scenarios that strictly require deterministic output, the random number generator used for random rewiring needs to be made deterministic with respect to the input graph. For example, the random number generator can be seeded with a hash of the input graph.

## 5.2 REPRODUCIBILITY

The source code of GRASS is available at https://github.com/grass-gnn/grass.

## ACKNOWLEDGMENTS

This work is partially supported by the NSF Award 2307698. We sincerely thank Martin Ritzert, Danica Sutherland, Hamed Shirzad, Hanke Chen, Owen Li, and Liangyuan Chen for their valuable feedback.

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

## A  RANDOM REWIRING

### A.1  MOTIVATIONS FOR SUPERIMPOSING RANDOM REGULAR GRAPHS

**Effects on Diameter.**  The diameter of a graph upper-bounds the distance between two nodes, and thus the number of layers for an MPNN to propagate information between them (Alon and Yahav, 2020). Superimposing a random regular graph on the input graph can drastically decrease its diameter. The least integer $d$ that satisfies

$$(r-1)^{d-1} \geq (2+\varepsilon)r|V|\log|V| \tag{18}$$

is the upper bound of the diameter of almost every random $r$-regular graph with $|V|$ nodes, where $r \geq 3$ and $\varepsilon > 0$ (Bollobás and Fernandez de la Vega, 1982). Since adding edges to a graph never increases its diameter, the diameter $d$ of the rewired graph is asymptotically upper-bounded by $d \in O(\log_r |V|)$ when $r \geq 3$. Subsequently, all nodes would be able to communicate with each other given $O(\log_r |V|)$ message passing layers, which could significantly reduce the risk of underreaching on large graphs. In addition, the diameter of a graph is a trivial upper bound of its effective resistance (Ellens et al., 2011), which has been shown to be positively associated with oversquashing (Black et al., 2023). Intuitively, it upper bounds the "length" of the bottleneck through which messages are passed.

**Effects on Internally Disjoint Paths.**  Since oversquashing can be attributed to squeezing too many messages through the fixed-size feature vector of a node (Alon and Yahav, 2020), increasing the number of internally disjoint paths between two nodes may reduce oversquashing by allowing information to propagate through more nodes in parallel. Intuitively, it increases the "width" of the bottleneck. A random $r$-regular graph with $r \geq 2$ almost certainly has a vertex connectivity of $r$ as $|V| \to \infty$ (Ellis, 2011). Menger's Theorem then lower-bounds the number of internally disjoint paths by a graph's vertex connectivity (Göring, 2000).

**Effects on Spectral Gap.**  Oversquashing has been shown to decrease as the spectral gap of a graph increases, which is defined as $\lambda_1$, the smallest positive eigenvalue of the graph's Laplacian matrix (Karhadkar et al., 2022). It has been proven that a random $r$-regular graph sampled uniformly from the set of all $r$-regular graphs with $|V|$ nodes almost certainly has $\mu < 2\sqrt{r-1} + 1$ as $|V| \to \infty$, where $\mu$ is the largest absolute value of nontrivial eigenvalues of its adjacency matrix (Puder, 2015). Since the graph is $r$-regular, its $i$-th adjacency matrix eigenvalue $\mu_i$ and $i$-th Laplacian matrix eigenvalue $\lambda_i$ satisfy $\lambda_i = r - \mu_i$ (Lutzeyer and Walden, 2017), lower-bounding the spectral gap with $\lambda_1 > r - 2\sqrt{r-1} - 1$.

### A.2  PSEUDOCODE OF THE PERMUTATION MODEL

---

**Algorithm 1** The Permutation Model (Friedman et al., 1989)

---

1: **procedure** PERMUTATIONMODEL($r$, $|V|$)
2:     $\sigma \leftarrow$ 2D array of size $(r, |V|)$
3:     **for** $i \leftarrow 0$ to $r - 1$ **do**
4:         $\sigma[i, :] \leftarrow$ RANDPERM($|V|$)    ▷ Random permutation of integers between 0 and $|V| - 1$
5:     **end for**
6:     $A \leftarrow$ Array of size $r * |V|$                                     ▷ Create an empty adjacency list
7:     **for** $j \leftarrow 0$ to $|V| - 1$ **do**
8:         **for** $k \leftarrow 0$ to $r - 1$ **do**
9:             $A[j * r + k] \leftarrow \{j, \sigma[k, j]\}$                    ▷ Add an edge to the adjacency list
10:         **end for**
11:     **end for**
12:     $A \leftarrow$ REMOVESELFLOOP($A$)                  ▷ Remove self-loops from the adjacency list
13:     $A \leftarrow$ REMOVEMULTIEDGE($A$)              ▷ Remove multi-edges from the adjacency list
14:     **return** $A$
15: **end procedure**

---

## B  GRASS ATTENTION

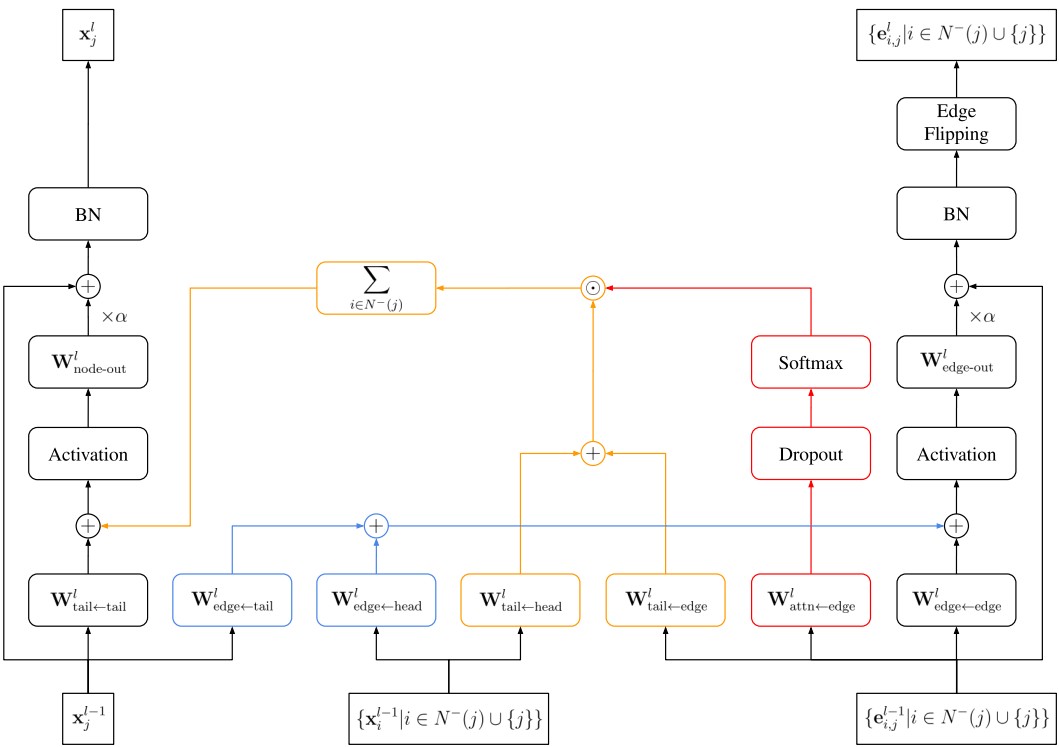

Figure 5: The structure of an attention layer of GRASS. Node aggregation is attentive, with attention weights derived from edge representations. Edge aggregation is done through an MLP. For simplicity, biases are not shown here.

## C  RESULTS ON HETEROPHILIC GRAPHS

Although tackling heterophily is not the main focus of this work, we have benchmarked GRASS on roman-empire (Platonov et al., 2023), a heterophilic graph, with results shown in Table 5. The experimental setup is identical to that described in Section 4.1, and hyperparameters can be found in Appendix F.2.

Table 5: Performance on the roman-empire dataset. The performance of GRASS shown here is the mean ± s.d. of 8 runs. The **best** and **second-best** results are highlighted. Performance numbers other than that of GRASS are adapted from Chen et al. (2024).

| Model | roman-empire Accuracy ↑ |
|---|---|
| GCN | $73.69 \pm 0.74$ |
| GAT (-sep) | $88.75 \pm 0.41$ |
| GPS | $82.00 \pm 0.61$ |
| NAGphormer | $74.34 \pm 0.77$ |
| Exphormer | $89.03 \pm 0.37$ |
| Polynormer | $\mathbf{92.55 \pm 0.37}$ |
| NeuralWalker | $\mathbf{92.92 \pm 0.36}$ |
| GRASS (ours) | $91.34 \pm 0.22$ |

# D    ADDITIONAL ABLATION STUDY RESULTS

Table 6: Ablation study results for the number of added edges per node on PascalVOC-SP. Reported values, except peak VRAM consumption, are the mean ± s.d. over 8 runs. The experimental setup is identical to that described in Table 3. For comparison, the variance of model performance due to randomness in the training process is 2.36e-5.

| # Added Edge per Node | 0 | 3 | 6 | 9 | 12 |
|---|---|---|---|---|---|
| Macro F1 ↑ | 0.4430 ± 0.0105 | 0.5606 ± 0.0102 | 0.5670 ± 0.0049 | 0.5612 ± 0.0056 | 0.5619 ± 0.0075 |
| Variance in Macro F1 Due to Random Rewiring | Deterministic | 3.40e-6 ± 2.12e-6 | 2.86e-6 ± 8.51e-7 | 2.27e-6 ± 9.32e-7 | 2.54e-6 ± 1.37e-6 |
| Training Time per Epoch (s) | 15.66 ± 0.07 | 29.55 ± 0.16 | 42.96 ± 0.09 | 56.31 ± 0.16 | 69.87 ± 0.28 |
| Peak VRAM (MiB) | 8525 | 14191 | 19893 | 25591 | 31231 |

Table 7: Ablation study results for graph encoding, graph rewiring, and attention mechanism on PascalVOC-SP. This table shows the performance of each ablated model as the mean ± s.d. over 4 runs. The experimental setup is identical to that described in Table 4. [*]Using RRWP encoding with 128 random walk steps would result in out-of-memory during preprocessing. With D-RRWP (128 eigenpairs, 128 steps), the preprocessed dataset has size 6.40 GiB, while with RRWP (32 steps), the preprocessed dataset has size 113.46 GiB, which is 17.73× larger.

| Setup | Macro F1 ↑ |
|---|---|
| GRASS | 0.5670 ± 0.0049 |
| D-RRWP (128 eigenpairs, 128 steps) → RRWP (32 steps)[*] | 0.5690 ± 0.0045 |
| D-RRWP (128 eigenpairs, 128 steps) → LapPE (128 eigenvectors) | 0.5387 ± 0.0070 |
| Random regular rewiring → Random non-regular rewiring | 0.5622 ± 0.0081 |
| GRASS attention → GAT attention (Veličković et al., 2017) | 0.4663 ± 0.0079 |
| GRASS attention → GatedGCN attention[†] (Bresson and Laurent, 2017) | 0.4414 ± 0.0075 |
| GRASS attention → Transformer attention[†] (Vaswani et al., 2017) | 0.4835 ± 0.0062 |

# E    COMPUTATIONAL PERFORMANCE

Table 8: Computational performance of GRASS on GNN Benchmark Datasets. Training time per epoch is the wall-clock time taken to complete a single training epoch, shown as the mean ± s.d. over 30 epochs. Preprocessing time is the wall-clock time taken to load, preprocess, and store the whole dataset prior to training. Specifications of the hardware used to run these experiments are also shown here.

| Dataset | ZINC | MNIST | CIFAR10 | PATTERN | CLUSTER |
|---|---|---|---|---|---|
| Training Time per Epoch (s) | 1.87 ± 0.07 | 11.83 ± 0.21 | 15.56 ± 0.10 | 33.58 ± 0.05 | 25.34 ± 0.03 |
| Preprocessing Time | 25s | 1m 1s | 1m 32s | 33s | 27s |
| Model Compilation | | | Yes | | |
| Activation Checkpointing | | | No | | |
| CPU | | | AMD Ryzen 9 9950X | | |
| GPU | | | NVIDIA RTX A6000 Ada | | |

Table 9: Computational performance of GRASS on LRGB datasets. Training time per epoch is shown as the mean ± s.d. over 30 epochs for Peptides-func and Peptides-struct, and over 10 epochs for PascalVOC-SP and COCO-SP. The definition of statistics are identical to that described in Table 8.

| Dataset | Peptides-func | Peptides-struct | PascalVOC-SP | COCO-SP |
|---|---|---|---|---|
| Training Time per Epoch (s) | 6.19 ± 0.30 | 5.90 ± 0.03 | 42.96 ± 0.09 | 539.50 ± 0.32 |
| Preprocessing Time | 1m 10s | 1m 32s | 3m 58s | 19m 49s |
| Model Compilation | | Yes | | Yes |
| Activation Checkpointing | | No | | Yes |
| CPU | | AMD Ryzen 9 9950X | | |
| GPU | | NVIDIA RTX A6000 Ada | | |

# F EXPERIMENTAL SETUP

## F.1 DATASETS

Table 10: Statistics of GNN Benchmark Datasets, adapted from Rampášek et al. (2022).

| Dataset | # Graphs | Avg. # Nodes | Avg. # Edges | Directionality | Task | Metric |
|---------|----------|--------------|--------------|----------------|------|--------|
| ZINC | 12000 | 23.2 | 24.9 | Undirected | Graph Regression | MAE ↓ |
| MNIST | 70000 | 70.6 | 564.5 | Directed | Graph Classification | Accuracy ↑ |
| CIFAR10 | 60000 | 117.6 | 941.1 | Directed | Graph Classification | Accuracy ↑ |
| PATTERN | 14000 | 118.9 | 3039.3 | Undirected | Node Classification | Accuracy ↑ |
| CLUSTER | 12000 | 117.2 | 2150.9 | Undirected | Node Classification | Accuracy ↑ |

Table 11: Statistics of LRGB datasets, adapted from Dwivedi et al. (2022).

| Dataset | # Graphs | Avg. # Nodes | Avg. # Edges | Avg. Short. Path | Avg. Diameter | Task | Metric |
|---------|----------|--------------|--------------|------------------|---------------|------|--------|
| Peptides-func | 15535 | 150.94 | 307.30 | 20.89 ± 9.79 | 56.99 ± 28.72 | Graph Classification | AP ↑ |
| Peptides-struct | 15535 | 150.94 | 307.30 | 20.89 ± 9.79 | 56.99 ± 28.72 | Graph Regression | MAE ↓ |
| PascalVOC-SP | 11355 | 479.40 | 2710.48 | 10.74 ± 0.51 | 27.62 ± 2.13 | Node Classification | Macro F1 ↑ |
| COCO-SP | 123286 | 476.88 | 2693.67 | 10.66 ± 0.55 | 27.39 ± 2.14 | Node Classification | Macro F1 ↑ |

## F.2 HYPERPARAMETERS

Table 12: Model hyperparameters for experiments on GNN Benchmark Datasets. Hidden layers of the task head, if any, use the GLU activation function (Dauphin et al., 2017).

| Model | ZINC | MNIST | CIFAR10 | PATTERN | CLUSTER |
|-------|------|-------|---------|---------|---------|
| # Parameters | 496545 | 103690 | 103738 | 495298 | 495558 |
| # Attention Layers | 49 | 15 | 15 | 53 | 53 |
| Attention Layer Dim. | 32 | 24 | 24 | 32 | 32 |
| Task Head Hidden Dim. | 192 | 144 | 144 | N/A (Linear) | N/A (Linear) |
| # Epochs | 2000 | 200 | 400 | 500 | 50 |
| Warmup Epoch Ratio | 0.1 | 0.05 | 0.1 | 0.1 | 0.1 |
| Batch Size | 200 | 200 | 200 | 200 | 200 |
| Initial Learning Rate | 1e-7 | 1e-7 | 1e-7 | 1e-7 | 1e-7 |
| Peak Learning Rate | 5e-4 | 1e-3 | 1e-3 | 1e-3 | 1e-3 |
| Final Learning Rate | 1e-7 | 1e-7 | 1e-7 | 3e-4 | 1e-7 |
| Betas | (0.95, 0.98) | (0.95, 0.98) | (0.95, 0.98) | (0.95, 0.98) | (0.95, 0.98) |
| Weight Decay Factor | 0.5 | 0.3 | 0.3 | 3.0 | 0.3 |
| Label Smoothing Factor | N/A (Regression) | 0.1 | 0.1 | 0.1 | 0.1 |
| Residual Connection Scale $\alpha$ | 0.2 | 0.4 | 0.4 | 0.2 | 0.2 |
| Random Walk Encoding Type | RRWP | RRWP | RRWP | RRWP | RRWP |
| (D-)RRWP Random Walk Length | 32 | 24 | 24 | 32 | 32 |
| Random Regular Graph Degree | 6 | 6 | 6 | 6 | 6 |
| Random Edge Removal Rate | 0.1 | 0.1 | 0.1 | 0.5 | 0.5 |

Table 13: Model hyperparameters for experiments on LRGB datasets and the roman-empire dataset. Hidden layers of the task head, if any, use the GLU activation function (Dauphin et al., 2017). *This dataset consists of a single graph.

| Model | Peptides-func | Peptides-struct | PascalVOC-SP | COCO-SP | roman-empire |
|-------|---------------|-----------------|--------------|---------|--------------|
| # Parameters | 500074 | 498315 | 501493 | 499377 | 2075730 |
| # Attention Layers | 48 | 48 | 53 | 53 | 24 |
| Attention Layer Dim. | 32 | 32 | 32 | 32 | 96 |
| Task Head Hidden Dim. | 192 | 192 | N/A (Linear) | N/A (Linear) | N/A (Linear) |
| # Epochs | 500 | 500 | 500 | 100 | 4000 |
| Warmup Epoch Ratio | 0.1 | 0.1 | 0.1 | 0.1 | 0.1 |
| Batch Size | 200 | 200 | 200 | 200 | 1* |
| Initial Learning Rate | 1e-7 | 1e-7 | 1e-7 | 1e-7 | 1e-7 |
| Peak Learning Rate | 1e-3 | 1e-3 | 1e-3 | 1e-3 | 1e-3 |
| Final Learning Rate | 1e-7 | 1e-7 | 1e-7 | 1e-7 | 1e-7 |
| Betas | (0.95, 0.98) | (0.95, 0.98) | (0.95, 0.98) | (0.95, 0.98) | (0.95, 0.98) |
| Weight Decay Factor | 0.3 | 3.0 | 1.0 | 0.3 | 1.0 |
| Label Smoothing Factor | 0.1 | N/A (Regression) | 0.1 | 0.1 | 0.1 |
| Residual Connection Scale $\alpha$ | 0.2 | 0.2 | 0.2 | 0.2 | 0.3 |
| Random Walk Encoding Type | D-RRWP | RRWP | D-RRWP | D-RRWP | D-RRWP |
| (D-)RRWP Random Walk Length | 128 | 64 | 128 | 64 | 256 |
| Random Regular Graph Degree | 3 | 3 | 6 | 6 | 3 |
| Random Edge Removal Rate | 0.1 | 0.1 | 0.1 | 0.1 | 0.5 |

