# OpenReview forum: "Greener GRASS: Enhancing GNNs with Encoding, Rewiring, and Attention"
_ICLR.cc/2025/Conference — ICLR 2025 Poster_

### Official Review · Reviewer_K5Pp · 2024-10-23

**Soundness:** 3
**Presentation:** 2
**Contribution:** 3
**Rating:** 6
**Confidence:** 4

**Summary:**

This paper proposes **GRASS**, a novel Graph Neural Network (GNN) architecture that combines **random walk encoding**, **graph rewiring**, and a new **graph-tailored additive attention mechanism**. The model introduces **Relative Random Walk Probabilities (RRWP)** encoding, a decomposed variant (**D-RRWP**) for computational efficiency, and uses **random regular graph rewiring** to enhance long-range communication between nodes. The paper demonstrates improved performance on several benchmark datasets, including a 20.3% lower MAE in ZINC. Ablation studies are conducted to analyze the contribution of each model component.

**Strengths:**

- **Novel combination**: The integration of RRWP encoding, random rewiring, and graph-tailored attention is innovative.
- **Performance**: The model shows strong results across multiple datasets, particularly in tasks requiring long-range dependencies.
- **Ablation studies**: Detailed ablation studies demonstrate the impact of each component on the overall model performance.

**Weaknesses:**

- **Clarity in methodology**: The explanation of RRWP and D-RRWP is overly detailed, with an abundance of variable names and step-by-step descriptions. To improve readability, consider starting with a high-level description of the algorithm's purpose and overall process before diving into specifics. Highlight one or two core equations that encapsulate the main idea, and move the more detailed steps to an appendix or illustrate them in a diagram. This approach would make the explanation more accessible while still providing depth for those interested in the finer details.
- **Scalability concerns**: While the authors claim good scalability, there are no experiments to validate this on larger datasets. A more thorough exploration of scalability would improve the strength of the claims. Additionally, reporting runtime and memory usage as graph size increases could offer valuable insights.
- **Dataset descriptions**: The datasets are not well-described, making it difficult for readers to understand the differences between them and why they were chosen. Including details such as the number of nodes and edges, task type, and dataset relevance to real-world applications would provide useful context.
- **Table presentation**: Table 1 contains unnecessary whitespace and could be more concise. Streamlining the table to present data more effectively without wasted space should be considered.

**Questions:**

- How does the model scale with larger datasets? Could you provide more empirical results to back the scalability claims?
- What motivated the choice of specific benchmark datasets? Were other datasets considered, particularly for tasks with very large graphs?
- Could you provide more details on the computational trade-offs of RRWP vs. D-RRWP?

---

### Official Review · Reviewer_z9x9 · 2024-10-31

**Soundness:** 3
**Presentation:** 3
**Contribution:** 1
**Rating:** 3
**Confidence:** 4

**Summary:**

This paper introduces GRASS, a framework that leverages the strengths of Graph Neural Networks (GNNs) and Graph Transformers (GTs), enhanced by D-RRWP encoding, random regular graph rewiring, and additive graph attention. Experimental results indicate that GRASS achieves competitive performance across several datasets.

**Strengths:**

- The paper is well-structured and accessible, with a clear presentation of methods and results.
- GRASS demonstrates superior performance in several benchmark datasets.

**Weaknesses:**

The framework builds upon established paradigms, leveraging techniques such as random walk encoding and random rewiring. Both techniques are recognized variants of prior methods [1][2], which raises questions about the extent of the framework's novelty. While this incorporation demonstrates methodological rigor, the contribution in terms of innovation appears constrained. It would be beneficial for the authors to clarify how their approach differs from or improves upon the methods in [1] and [2]. Are there specific aspects of their methodology or implementation that extend or address limitations in these prior works? Providing such explanations would strengthen the argument for the framework's originality.

In terms of performance, GRASS demonstrates marginal or minimal improvement on datasets like CLUSTER, Peptides-func, and Peptides-struct. The datasets used in the experiments are limited in scale and scope, primarily featuring small graphs with an average of 500 nodes per graph and focusing on domains such as molecular chemistry and computer vision. While these domains are valuable, the scope of evaluation is not comprehensive enough to fully assess the proposed GNN architecture. To address this limitation, the authors could extend their evaluation to larger and more diverse datasets, such as those in the OGB benchmark [3], which includes large-scale networks like ogbn-arxiv (citation) and ogbn-products (sales). Additionally, a justification for the choice of current datasets would help contextualize their experimental focus. If there are technical constraints that prevent evaluation on larger graphs, discussing these limitations could provide valuable insights into potential scalability challenges.

[1] Graph inductive biases in transformers without message passing.

[2] Exphormer: Sparse transformers for graphs.

[3] https://ogb.stanford.edu/

**Questions:**

The Random Rewiring module, which introduces random edges into the graph structure, raises a critical question: could this random addition disrupt the integrity of the original graph by introducing noise? This potential limitation warrants further investigation.

The attention mechanism in GRASS is designed to aid in selective aggregation. However, it remains unclear whether this feature significantly enhances GRASS's performance on heterophilous graphs, where connected nodes often have different labels. Providing empirical results or case studies to demonstrate the mechanism's effectiveness in such settings would strengthen the evaluation and highlight the framework's adaptability to varied graph types.

---

### Official Review · Reviewer_2g6k · 2024-11-04

**Soundness:** 3
**Presentation:** 3
**Contribution:** 2
**Rating:** 6
**Confidence:** 3

**Summary:**

The paper proposes combining random graph rewiring, RRWO encoding and a specially crafted additive attention mechanism. To improve the efficiency of RRWP encoding, the authors utilize the truncated eigen decomposition. The experimental results show favorable results for the proposed combination of the utilized techniques.

**Strengths:**

* Strong experimental results on various datasets.
* The paper is well written.
* Figures help communicate the ideas more clearly.

**Weaknesses:**

* Misses citation to [1], I think it should be discussed in related work and any similarities/difference should be made clear when it comes to the model architecture in [1] vs the author's proposed additive attention mechanism, which is claimed to be a novel contribution.

[1] https://www.nature.com/articles/s41598-023-44224-1

**Questions:**

* Have the authors seen [1], the edge attention scheme proposed by the authors looks similar to the model architecture used there. What is the difference between the proposed attention mechanism with the model architecture in [1]?
* What is the advantage of generating random regular graphs in the proposed way when each node can sample nodes from the whole graph without replacement in a simpler manner?
* Can the authors also present results for the attention mechanism part alone, adding encodings to an existing GNN model is possible most of the time so it would be valuable if Tables 1 and 2 contained results for variants of GRASS not including random rewiring and
RRWP encoding.

[1] https://www.nature.com/articles/s41598-023-44224-1

---

### Official Review · Reviewer_budq · 2024-11-04

**Soundness:** 3
**Presentation:** 3
**Contribution:** 2
**Rating:** 5
**Confidence:** 4

**Summary:**

This paper introduces GRASS (Graph Attention with Stochastic Structures), a new GNN architecture combining three key elements: graph encoding through Relative Random Walk Probabilities (RRWP), random rewiring of graph structures, and a novel additive attention mechanism. GRASS aims to improve long-range dependencies and selective node aggregation. The authors demonstrate that GRASS outperforms baseline models across several benchmark datasets and supports scalable GNN performance.

**Strengths:**

- The paper is well written and easy to follow.
- The integration of RRWP encoding, random rewiring, and attention tailored to graph-structured data offers a plausible approach to addressing oversquashing and underreaching in GNNs.
- The proposed D-RRWP variant is a computationally efficient alternative to traditional RRWP encoding, making GRASS more feasible for large graphs.
- The results on Pascal-VOC and COCO-SP seems promising.

**Weaknesses:**

- Limited experimentation: Although GRASS is designed for scalability, the paper acknowledges that scalability testing on large, real-world graphs is limited due to resource constraints. Without large dataset evaluations, the real-world applicability to large-scale graph datasets remains unclear. For the currently used datasets, the memory or scalability does not present significant concerns since they are graph-level tasks with a small number of average nodes per graph. For example, the benchmarks are standard in GNN research, yet additional evaluations on real-world applications (e.g., social networks or knowledge graphs) could provide more insight into practical applicability.
- The proposed attention mechanism in GRASS leverages additive attention with edge features to capture relationships uniquely tailored for graph structures. However, random walk encodings like RRWP and related techniques such as Personalized PageRank (PPR) are already established in the graph learning community, as are various graph rewiring methods. To strengthen the paper’s contribution, it would be helpful if the authors could clarify how their additive attention mechanism distinctly advances existing methods.

**Questions:**

- See weaknesses.
- It seems like the recent work NeuralWalker has made significant improvements in comparison to prior baselines. Can you clarify its significance and similarities to your work?
- Can you clarify your procedure for hyper-parameters? Te recent work [1] suggests that many of the baselines for LRGB can be easily improved by simple hyper-parameter changes.
- What impact does the frequency of rewiring during training have on model performance and training stability? Have experiments been conducted on different rewiring frequencies to understand this effect?

[1] Where Did the Gap Go? Reassessing the Long-Range Graph Benchmark

---

### Official Review · Reviewer_eNkJ · 2024-11-05

**Soundness:** 3
**Presentation:** 3
**Contribution:** 2
**Rating:** 6
**Confidence:** 3

**Summary:**

This paper proposes a random-walk-based approach to encode structural information, and rewire the graph accordingly, to enhance the performance of attention-based gnns.

**Strengths:**

The motivation behind this paper is clear. The method justification is backed by evidence.
Many types of baselines and ablation studies are considered when evaluating the model performance.
The design goals are clear.

**Weaknesses:**

A lot of baseline performance numbers in tables are missing. I'm not sure if there are issues with running these models with the baselines.

There are many aspects that could affect model design for given graph data. For example, homophily / heterophily properties; sparsity; clustering properties etc. could result in different model performances across designs. It's not clear what would be the type of data where the proposed method can achieve improvements.

The algorithm description could benefit from more clarity.

**Questions:**

The random-walk-based encoding technique bears a lot of similarity to the position / structural encodings in the context of graph transformer (e.g. random walk encoding in https://arxiv.org/abs/2110.07875). What are the key differences and similarity? How would you demonstrate the advantage of the proposed algorithm in light of those related works?
Since strategies like random walk encoding, random graph rewiring, and attention mechanisms are commonly employed in graph domains, the authors should emphasize the novelty of their framework.

It would be beneficial to explore how GRASS performs on other graph-related tasks, such as link prediction or edge classification. What would be the type of tasks where the proposed method would achieve an advantage?

---

### Meta-Review · Area_Chair_iUif · 2024-12-20

**Metareview:**

This paper introduces GRASS, a Graph Neural Network (GNN) architecture that integrates Relative Random Walk Probabilities (RRWP) encoding, random regular graph rewiring, and a new graph-tailored additive attention mechanism. GRASS aims to enhance long-range dependencies and selective node aggregation in graph models. Experimental results demonstrate that GRASS outperforms baseline models across several benchmark datasets.

The paper is well-written and presents a clear motivation for combining RRWP encoding, random rewiring, and an additive attention mechanism. The integration of these elements is innovative and shows promise in addressing issues like oversquashing and underreaching in GNNs.

However, the technical novelty is somewhat limited, as it builds upon established techniques without sufficiently clarifying how it advances beyond existing works. Important citations and comparisons to closely related methods are missing or improperly cited, particularly concerning the structure of an attention layer and attention mechanisms and encoding strategies. The scalability claims are unvalidated due to limited experimentation on large-scale, real-world graphs, raising concerns about the model's applicability in practical settings.

Overall, based on the evaluations from reviewers, I recommend the acceptance of this paper. The authors are encouraged to improve the paper quality by consulting related survey papers to accurately identify the original sources of certain model components, highlighting the novel contributions over prior work, providing thorough comparisons and validating scalability on larger graphs.

**Additional Comments On Reviewer Discussion:**

During the rebuttal period, the authors actively addressed the reviewers' concerns, leading to two reviewers raising their scores. However, Reviewer z9x9, who initially gave a score of 3, remained inactive throughout the entire rebuttal period. Therefore, the AC lower the Reviewer z9x9's confidence when making the final decision.

---

### Decision · Program_Chairs · 2025-01-22

Accept (Poster)